# Advances in the Mechanistic Understanding of Iron Oxide Nanoparticles’ Radiosensitizing Properties

**DOI:** 10.3390/nano13010201

**Published:** 2023-01-02

**Authors:** Indiana Ternad, Sebastien Penninckx, Valentin Lecomte, Thomas Vangijzegem, Louise Conrard, Stéphane Lucas, Anne-Catherine Heuskin, Carine Michiels, Robert N. Muller, Dimitri Stanicki, Sophie Laurent

**Affiliations:** 1General, Organic and Biomedical Chemistry Unit, NMR and Molecular Imaging Laboratory, University of Mons (UMONS), B-7000 Mons, Belgium; 2Medical Physics Department, Institut Jules Bordet, Université Libre de Bruxelles (ULB), B-1070 Brussels, Belgium; 3Center for Microscopy and Molecular Imaging (CMMI), B-6041 Gosselies, Belgium; 4Namur Research Institute for Life Sciences (NARILIS), University of Namur, B-5000 Namur, Belgium

**Keywords:** iron oxide nanoparticles, X-ray irradiation, radiosensitization, biological mechanism, thioredoxin reductase, cancer therapy

## Abstract

Among the plethora of nanosystems used in the field of theranostics, iron oxide nanoparticles (IONPs) occupy a central place because of their biocompatibility and magnetic properties. In this study, we highlight the radiosensitizing effect of two IONPs formulations (namely 7 nm carboxylated IONPs and PEG_5000_-IONPs) on A549 lung carcinoma cells when exposed to 225 kV X-rays after 6 h, 24 h and 48 h incubation. The hypothesis that nanoparticles exhibit their radiosensitizing effect by weakening cells through the inhibition of detoxification enzymes was evidenced by thioredoxin reductase activity monitoring. In particular, a good correlation between the amplification effect at 2 Gy and the residual activity of thioredoxin reductase was observed, which is consistent with previous observations made for gold nanoparticles (NPs). This emphasizes that NP-induced radiosensitization does not result solely from physical phenomena but also results from biological events.

## 1. Introduction

Cancers remain a major threat to human health and quality of life. After surgery, radiotherapy (RT) is considered the primary modality for cancer treatment and it is estimated that more than 50% of patients will receive it either alone or in combination with other treatments [1,2]. Despite major developments (e.g., hypofractionated and high-dose-rate FLASH RT treatments), RT still has limitations in that a dose is received by healthy tissues surrounding the tumor, which is responsible for the induction of side effects, including in some cases radiation-induced cancers [3,4,5]. Over the last decades, the main advances in the field have focused on the implementation of advanced radiation delivery techniques, such as intensity-modulated radiotherapy (IMRT) or the use of charged particles instead of classical X-rays [6]. Although these advances improve the dose conformity and the clinical outcome, additional therapeutic benefits may be gained by using RT combined with “radiosensitizers”, e.g., molecules or particles that enable access to a given tumor cell, allowing for its destruction using a reduced total dose delivered to the patient. Specifically loading these molecules into the tumor would therefore allow the total radiation dose delivered to the patient to be decreased; thus, reducing the probability of normal tissue complications [7]. Driven by the pioneering work of Hainfeld et al. [8] and the recent developments in nanomedicine, many in vitro and in vivo studies have demonstrated the ability of nanoparticles, when injected into the tumor, to amplify RT efficiency. Despite a strong interest in this field of investigation, many studies only focus on high-Z particles (typically gold, platinum, gadolinium, etc.) because of their higher X-ray absorption cross-section [9,10]. Indeed, in nanoparticle-based radio-enhancement, secondary emission (photons and electrons) induced by nanoparticles during irradiation triggers the generation of reactive oxygen species (ROS) in the vicinity of the nanoparticles (NPs). Depending on the nanoparticle subcellular localization, these reactive species can subsequently damage DNA, membranes and other essential cellular components. Therapeutic efficiency therefore depends on a multitude of parameters, including the physicochemical characteristics of the nanoparticles, their subcellular localization, as well as cell type-specific properties (e.g., DNA repair capacity, antioxidant expression level, etc.) [11]. 

While some studies demonstrated promising results for hybrid nanomaterials [12], ref. [13] (typically gold/iron oxide in Janus or core/shell configurations), the use of iron oxide nanoparticles (IONPs) as dose enhancers remains poorly studied, with recent reports mostly focusing on the combination of RT and magnetic hyperthermia [14]. However, some other studies have highlighted the additive or synergistic effects on cell death when using IONPs in combination with either X-rays or particle beams [15,16] as a result of their ability to produce ROS through an iron-catalyzed Fenton/Haber–Weiss reaction or to induce increased DNA damage [17,18,19]. It should also be mentioned that ROS-independent mechanisms have been recently proposed through the ability of IONPs to interact with HSP90, which is a key protein regulating the mitotic process as well as DNA repair [20]. 

These results appear particularly promising as these agents are already commonly used in the biomedical field [21]. One has to remember that IONPs (more specifically maghemite and magnetite) have been widely used as contrast enhancers for clinical magnetic resonance imaging (MRI), as a result of their biocompatibility and remarkable (super)paramagnetic properties [22,23]. Since image-guided RT based on MRI workflow is becoming clinically available, there is a clinical need to evaluate the radiosensitizing properties of this potential theranostic agent [24].

In the field of oncology, passive targeting of solid tumors using nanoparticles depends on the proposed vascular permeability of the tumor microenvironment (i.e., by the enhanced permeability and retention (EPR) effect). Although this effect remains controversial and is highly dependent on the tumor type, it can be hypothesized that a higher accumulation should be achieved for particles able to circulate for a long time in the bloodstream [25]. Previous studies have described the excellent properties achieved by using a polyethylene glycol (PEG)-based coating, which helps to reduce the non-specific adsorption of blood proteins and acts as a protective layer with respect to phagocyte adhesion [26,27]. In work from our group, we have demonstrated, using a multimodal approach, that longer circulation times are reached for longer PEG chains (i.e., PEG_5000_). Interestingly, when it comes to the study of the radiosensitization effects induced by magnetic nanoparticles, most reports focus on negatively charged systems (i.e., citric [17] or dimercaptosuccinic acids [14]) presenting varying degrees of agglomeration. For all these reasons, we were interested in focusing on PEGylated monocore IONPs in the context of proton and photon radiation therapy and comparing their behavior to that obtained by negatively charged particles (i.e., carboxylated IONPs). The potential link existing between the amount of internalized particles (by modulating the exposure time and modifying their surface) and the radiobiological effects was evaluated in vitro in A549 cells using 225 kV X-rays. According to some recent reports, some authors have suggested that the contribution of biochemical mechanisms were complementary to the physical ones. More specifically, the thioredoxin reductase (TrxR) enzyme was identified as a new potential biological target [28,29,30]. For these reasons, the ability of the above-mentioned nanosystems to inhibit the TrxR activity is proposed. A potential correlation with the determined radiosensitization effect was also hypothesized. 

## 2. Material and Methods

### 2.1. Materials

Ferric chloride solution (FeCl_3_, 45%), ferrous chloride tetrahydrate (FeCl_2_.4H_2_O, >99%) and sodium hydroxide were purchased from Fluka (Bruxelles, Belgium). The compound 3-(triethoxysilyl)propyl succinic anhydride (TEPSA) was purchased from ABCR (Karlsruhe, Germany). *N*-(3-dimethylaminopropyl)-*N*’-ethylcarbodiimide hydrochloride (EDC), citric acid, aprotinin, (3-(4,5-dimethylthiazol-2-yl)-2,5-diphenyltetrazolium bromide) (MTT reagent), calcium chloride dihydrate, sodium chloride, magnesium chloride, sodium citrate dihydrate, diethylene glycol (DEG), dimethyl sulfoxide (DMSO), glycerol, *N*-Boc-1,4-butanediamine, thioredoxin reductase assay kit, sodium lactate, sodium pyruvate, sodium phosphate dibasic heptahydrate, sodium tartrate dihydrate, dimethylformamide (DMF), acetone and diethyl ether were purchased from Sigma-Aldrich (Overijse, Belgium). Potassium ferrocyanide, iron standard solution 1000 μg·mL^−1^, sodium sulfate and formaldehyde 37% were purchased from VWR (Leuven, Belgium). α-Methoxy-ω-amino poly(ethylene glycol) 5000 was purchased from Iris Biotech GMBH (Marktredwitz, Germany). Trypan blue (4%), minimum essential medium (MEM) GlutaMAX supplement, phosphate-buffered saline, Pierce 660 nm protein assay reagent, fetal bovine serum (FBS), penicillin-streptomycin (10,000 U/mL) and LysoTracker blue DND-22 were purchased from Thermo Fischer (Merelbeke, Belgium). Lissamine rhodamine B sulfonyl chloride was purchased from Acros Organics (Geel, Belgium).

### 2.2. Iron Oxide Nanoparticles Synthesis (IONPs)

IONPs were synthesized following a previously described process [31,32]. A detailed procedure is described in the supporting information.

### 2.3. Characterization of IONPs

The measurements of the size distribution and the ζ-potential of the nanoparticles suspended in aqueous media were performed by photon correlation spectroscopy (PCS) with a Zetasizer nano zs (Malvern Instruments, Malver, UK) using a He–Ne laser (633 nm). The ζ-potential was determined directly in a solution containing NaCl (0.01 mM). The pH of the aqueous solution containing NPs was adjusted by adding 0.1–0.001 mM HNO_3_ or NaOH solution. Transmission electron microscopy (TEM) was used to obtain detailed information about the morphology and size of the samples and was carried out using a Fei Tecnai 10 microscope (Hillsboro, Oregon, USA) operating at an accelerating voltage of 80 kV. The samples were prepared by placing a drop of diluted suspension on a copper-grid (300 mesh), allowing the liquid to dry in air at room temperature. The statistical analysis of the TEM images was performed by iTEM (Münster, Germany) on multiple images for each sample. The mean diameter, standard deviation (SD) and polydispersity index (PDI) were calculated by measuring the particle diameter on statistically significant samples (from 500 to 700 particles). Longitudinal (R_1_) and transverse (R_2_) relaxation rate measurements at 0.47 T were obtained on a Minispec mq 20 spin analyzer (Bruker, Karlsruhe, Germany). The relaxation rates were measured as a function of the iron molar concentration at 0.47 T in order to calculate the *r*_1_ and *r*_2_ relaxivities (defined as the enhancement of the water proton relaxation rate in 1 mmol.L^−1^ solution of contrast agent). The relaxivities were calculated as the slope of relaxation rate (R_*i*_^*obs*^) versus iron concentration according to the equation: (1)$$R_{i}^{obs}=\frac{1}{T_{i}^{obs}\;}=r_{i}\left[Fe\right]+\frac{1}{T_{i}^{dia}}$$
where *r_i_* are the relaxivities and $T_{i}^{dia}$ are the proton relaxation times in aqueous solutions without nanoparticles. The X-ray diffraction (XRD) experiment was performed on a D5000 Siemens diffractometer (Munich, Germany) using the Cu K_a_ radiation (l = 0.154056 nm). The scattering intensities were measured over an angular range of 0 < Q < 120 for all samples with a step size (2Q) of 0.05° and a step time of 25 s.

Infrared spectra of dried samples were recorded on an FTIR spectrometer. The measurements were performed on a Perkin Elmer Spectrum 100 spectrometer (UK) using the solid potassium bromide method, with 2 cm^−1^ resolution and 10 scans.

Thermogravimetric analyses (TGA) were performed on a TA Q5000 system (TA Instruments, USA). The mass loss of the pre-dried samples (80 °C during 24 h) was monitored under nitrogen from room temperature to 120 °C at a heating rate of 10 °C·min^−1^. After an isotherm at 120 °C under nitrogen for 10 min to remove the bound water, the samples were heated from 120 °C to 600 °C at a heating rate of 10 °C·min^−1^ under air.

The organic and inorganic contents were determined from the weight loss on the TGA curves (Appendix A). We extracted the number, *n_TEPSA_*, of organosilane molecules grafted to IONPs from Equation (2) [33]: (2)$$\frac{org.\;wt.\;\%}{inorg.\;wt.\;\%}=\frac{n_{TEPSA}\times M_{w}^{TEPSA}}{M_{w}^{IONP}}$$
where $M_{w}^{IONP}$ is the molar mass of the particle estimated by: (3)$$M_{w}^{IONP}=\frac{4}{3\;}\;\pi \;R^{3}\times N_{A}\times \rho$$
where *N_A_* is Avogadro’s number, *ρ* the mass density of Fe_3_O_4_ = 5180 kg.m^–3^ and *R* is the particle radius (3.5 nm determined from TEM analysis). For IONPs firstly coated with TEPSA then coupled to polymer chains, the organic content measured by TGA is the sum of two components: (4)$$\frac{org.\;wt.\;\%}{inorg.\;wt.\;\%}=\frac{n_{TEPSA}\times M_{w}^{TEPSA}}{M_{w}^{IONP}}+\frac{n_{polymer}\times M_{w}^{polymer}}{M_{w}^{IONP}}$$

As the grafting density of the silane should not be affected by the coupling reaction with the polymer, the weight ratio of TEPSA relative to IONP was considered constant for all samples. 

### 2.4. Degradation of IONPs in Artificial Lysosomal Fluid (ALF) and in Simulated Body Fluid (SBF)

ALF and BLF were prepared according to Rabel et al. [34]. After membrane filtration (membrane cut-off: 14 kDa), the sterilized fluids were stored at 4 °C.

For degradation studies, IONPs were diluted in ALF or SBF media (final [Fe] = 5 mM) and then incubated at 37 °C with magnetic stirring. After given times (3, 6, 13, 17, 20, 24, 37, 42, and 48 h), the concentration of elemental iron released from the degradation of IONPs was determined from the filtrate collected in ultracentrifugation cells after 15 min (7.250 G). As described by Boutry et al. [35]*,* the Prussian blue colorimetric method was used to determine the concentration of iron calculated from the calibration curve made from a known concentration of iron standard (iron standard 1000 µg·mL^−1^ (VWR, Belgium), diluted in Milli-Q). The mineralized solution (100 μL) was mixed with 100 μL of a 5 M HCl solution and 100 μL 5% potassium ferrocyanide solution (VWR, Belgium), and absorbance was measured at a wavelength of 650 nm using microplate readers (SpectraMax).

The degradation process was described by Gompertz fitting defined as: (5)$$y=y_{m}*{\left(\frac{y_{0}}{y_{m}}\right)}^{e^{\left(-Kx\right)}}$$
where $y_{m}$ describes the maximum percentage of degradation of IONPs. $y_{0}$ is the starting percentage of IONPs degradation, K is the degradation rate of IONPs and x is the time.

### 2.5. Cell Culture

Human lung carcinoma A549 cells were cultured in Eagle’s Minimum Essential Medium (MEM Glutamax; Gibco by Life Technologies, Boston, MA, USA) supplemented with 10% of fetal bovine serum (FBS; Gibco by Life Technologies, MA, USA) and 1% of Pen-strep in a humidified atmosphere incubator containing 5% CO_2_ at 37 °C. 

### 2.6. Cell Viability 

The cellular metabolic activity in the presence of our homemade nanoparticles was determined by 3-[4,5-dimethylthiazol-2-yl]-3,5 diphenyl tetrazolium bromide (Merk KGaA, Darmstadt, Germany) (MTT method). A549 cells were seeded at 5 × 10^4^ cells/well in 96-well plates (Greiner, Wemmel, Belgium) and placed in the incubator at 37°C with 5% CO_2_ overnight. Then, the cells were incubated for 24 or 48 h with an increased concentration of freshly prepared iron oxide nanoparticles dispersions ([Fe] = 0, 10, 25, 50, 100, 150 and 200 μg·mL^−1^). After the given incubation time, the medium was removed, and the wells were filled with the MTT solution (500 μg·mL^−1^ in PBS). After 3 h of incubation at 37 °C, the MTT solution was removed, formazan crystals were dissolved with 100 μL of DMSO and the absorbance was measured at 570 nm using a microplate reader (SpectraMax M2, Molecular Devices, CA, USA). Cell viability was determined by the comparison between the percentage of absorption of untreated and treated cells. 

### 2.7. Iron Dosing Method by Perls’ Prussian Blue Reaction

A549 cells were seeded at a concentration of 3 × 10^5^ cells/well in 6-well plates (Greiner, Wemmel, Belgium) and placed in an incubator at 37 °C with 5% CO_2_ overnight. The medium was then replaced by MEM supplemented with 10% of FBS with ([Fe] = 50 μg·mL^−1^) or without IONPs. After twenty-four hours, the cells were washed three times with PBS, trypsinized and pelleted by centrifugation (1000 rpm, 5 min, 25 °C). After counting the number of cells, cell pellets were mineralized in 200 μL of 5 M HCl for 72 h in a dry block heater at 37 °C. Subsequently, the mineralized solution was mixed with 5% potassium ferrocyanide solution (VWR, Belgium) and the absorbance was measured at the wavelength of 650 nm using microplate readers (SpectraMax M2, Molecular Devices, CA, USA). 

As already described, iron content was expressed as pg Fe/cell, calculated from the calibration curve made from a known concentration of the iron standard (VWR, Belgium). 

### 2.8. Nanoparticle Localization 

Particle uptake in A549 cells was analyzed by confocal laser scanning microscopy. A549 cells were seeded at a concentration of 1.1 × 10^5^ cells/well in μ-slide 4-well ibiTreated (Proxylab, Beloeil, Belgium) and placed in an incubator at 37 °C with 5% CO_2_ overnight. The medium was then replaced by MEM supplemented with 10% of FBS with ([Fe] = 50 µg·mL^−1^) or without of rhodamine-labeled IONPs for 2 h, 6 h, 24 h and 48 h. Finally, the cells were incubated for 30 min with the LysoTracker Blue at a final concentration of 50 nM in the growth medium without NPs. Cells were examined with a Zeiss LSM710/AxioObserver Z1 confocal microscope using a plan-Apochromat 63x/NA 1.4 oil DIC M27 immersion objective in a thermostatized chamber (XL/LSM incubator, Zeiss; Tempcontrol 37-2, PeCon) at 37 °C. The images of the samples were acquired in z-stack (15 slices) mode with a slice thickness of 0.68 µm.

### 2.9. Thioredoxin Reductase (TrxR) Activity 

The TrxR activity was evaluated by a commercial kit (Merk KGaA, Darmstadt, Germany). A549 cells were seeded at a concentration of 1.26 × 10^6^ cells in T25 flasks (VWR, PA, USA) and placed in an incubator at 37 °C with 5% CO_2_ overnight. The media were then replaced by MEM supplemented with 10% of FBS with IONPs ([Fe] = 50 μg·mL^−1^) and without for the control cells. After twenty-four hours, the cells were rinsed three times with PBS, trypsinized and pelleted by centrifugation (1000 rpm, 5 min, 4 °C). Cells were resuspended in a lysis buffer (9% *w*/*w* sucrose; 5% *v*/*v* aprotinin (Merk KGaA, Darmstadt, Germany) in deionized water) and lysed by a Dounce homogenizer. The catalytic reduction of 5,5-dithiobis(2-nitrobenzoic) acid (DTNB) to 5-thio-2-nitrobenzoic acid (TNB) by TrxR was followed by recording the absorbance at 412 nm for 10 min using a spectrophotometer (SpectraMax M2, Molecular Devices, Sacramento, CA, USA). 

### 2.10. X-ray Irradiation 

The X-ray irradiation response was determined with a homogenous X-ray beam produced by an X-Rad 225 XL (PXi Precision X-ray, North Branford, CT, USA) at 225 kV with a dose rate of 2 Gy min^−1^. Forty-eight hours before irradiation, cells were seeded at a concentration of 5 × 10^4^ cells/wells in 24-well plates (Greiner, Wemmel, Belgium) as 50 μL drops. After 2h, the wells were then filled with MEM + 10% of FBS and placed in the incubator at 37 °C with 5% CO_2_ overnight. The medium was then replaced by MEM supplemented with 10% of FBS with ([Fe] = 50 μg·mL^−1^) or without IONPs and incubated at 37 °C for 24 h until irradiation. Before irradiation, the cells were rinsed with PBS to discard nanoparticles and the wells were filled with MEM supplemented with 10% of FBS. 

### 2.11. Proton Irradiation 

A XY spatially homogenous proton beam was produced by a 2 MV tandem accelerator (High Voltage Engineering Europa). Forty-eight hours prior the irradiation, cells were seeded as 32 μL drops at a concentration of 5 × 10^4^ cells in sterilized irradiation chambers [36]. Two hours after seeding, the chambers were filled with MEM supplemented with 10% of FBS and placed in the incubator at 37 °C with 5% CO_2_ overnight. Twenty-four hours before irradiation, the medium was then replaced by MEM supplemented with 10% of FBS with/without IONPs ([Fe] = 50 μg·mL^−1^). Before irradiation, the medium was removed, the chambers were rinsed with PBS and filled with medium supplemented with 10% of FBS. 

A beam energy of 1.3 MeV was selected corresponding to an average linear energy transfer of 25 keV µm^−1^ in the thickness of the cell layer (computed using SRIM software [37]). The dose rate was fixed at 2 Gy min^−1^ and the dose range was chosen to cover the survival fraction down to a few percent. All doses were calculated using the classic broad beam formula:(6)$$D=\frac{1.6\times 10^{-9}\times LET\times \phi }{\rho }$$
where the density *ρ* is taken as 1 g cm^–^³ and *φ* is the proton beam fluence (particles cm^–^²). 

### 2.12. Clonogenic Assay

Following irradiation, the cells were detached (using 0.25% trypsin) and counted after irradiation. The cells were seeded in 6-well plates containing MEM supplemented with 10% of FBS and 1% of penicillin/streptomycin to obtain countable colony numbers for different radiation doses. To obtain the precise number of cells seeded, cells were also seeded in 24-well plates, fixed with 4% paraformaldehyde (Merck Chemicals, Overijse, Belgium) 2 h after seeding, washed with PBS two times and finally counted manually under an optical microscope. Eleven days after irradiation, the colonies were stained with violet crystal in 2% ethanol. The colonies were counted manually and represented the surviving cells. For each irradiation dose, the plating efficiency (PE) was calculated as the ratio of the number of colonies by the initial number of seeded cells for each experimental condition. The surviving fraction (SF) was determined by dividing the PE of irradiated cells by the PE of control cells. To evaluate the radiosensitizing effect of IONPs, the sensitization enhancement ratio (SER) and the amplification factor (AF) were calculated. The AF reflects the enhancement of cell death in the presence of IONPs compared with radiation alone at a given dose, in this case, 2 Gy [38].
(7)$$SER=\frac{Radiation\;dose\;without\;IONPs\;}{Radiation\;dose\;with\;IONPs}$$
(8)$$AF\left[\%\right]=\frac{SF_{control}^{\;2Gy}-SF_{\;IONPs}^{\;2Gy}}{SF_{control}^{\;2Gy}}\times 100$$

### 2.13. Statistical Analysis

Statistical analysis was completed using Prism 8 (GraphPad Software, San Diego, CA, USA). All experiments were independent and repeated at least three times on separate days. A one-way ANOVA was used to compare the difference between groups and results were reported as means ± SD. The number of asterisks in the figures indicates the level of statistical significance as follows: * *p* < 0.05, ** *p* < 0.01, *** *p* < 0.001.

## 3. Results 

### 3.1. Nanoparticle Synthesis and Characterization 

The IONPs were prepared by alkaline co-precipitation of ferrous/ferric chlorides in polyol medium and stabilized by means of a silanization step using a carboxylated organosilane (i.e., TEPSA). As illustrated in Figure 1A–C, TEM images show the formation of spherical IONPs with a mean diameter of 7 nm. The X-ray diffraction experiments attested the phase of IONPs (Appendix A) (30.2°, 35.5°, 42.8°, 57.2° and 62.7°). Using the Scherrer equation, a particle size of 5.89 nm was estimated from the peak with the highest intensity (peak corresponding to the [311] plane at 2θ = 35.5°). Zeta potential measurements suggested the efficiency of the surface modification by the shift of the point of zero charge (PZC) from 6.7 for uncoated objects to 3.5 after treatment/purification (Figure 1D). Furthermore, an increase in the mean diameter from 18 nm (before coating) to 21 nm (after treatment) was observed by photon correlation spectroscopy (PCS) (Figure 1E). Finally, FTIR spectroscopy (Appendix A) highlighted the appearance of bands around 2900 cm^−1^ and 1710 cm^−1^, attributable to C–H and C=O bond vibrations, respectively, as well as bands in the region of 1100 cm^−1^ characteristic of Si–O–Si bond vibrations, which confirm the formation of a polysiloxane shell.

Starting from this batch, polyethylene glycol chains (i.e., PEG_5000_) were grafted through a classical EDC coupling process using α-amino and ω-methoxy-PEG. While no difference in the core size was noticed by TEM (Figure 1C), PCS indicated a slight increase in the mean hydrodynamic size from 21 for carboxylated IONPs to 28 nm for PEGylated ones (Figure 1E). The evolution of the zeta potential value at pH 7 from −37 mV to −4 mV after PEG treatment, confirmed the surface modification. 

### 3.2. Cell Uptake and Associated Toxicity 

Preliminary cytotoxicity investigations of IONPs on A549 cells were performed by MTT assay. No significant decrease in the cell viability was observed for cells preincubated for 6 h, 24 h or 48 h with carboxylated IONPs compared to untreated controls (Appendix A). In the case of PEGylated particles, while no significant toxicity was observed for cells preincubated for 6 and 24 h, a slight decrease in the cell viability (around 10%) was observed after 48 h when treated with quantities ranging from 50 µg to 200 µg·mL^−1^ of Fe. This result showed that the as-prepared IONPs had a negligible influence on the cell viability, as already suggested in other studies [39,40]. In the frame of this work, we established that an incubation of 50 μg·mL^−1^ of Fe in the medium did not generate any critical cytotoxicity (Appendix A), therefore allowing further internalization and irradiation experiments. Following these results, the IONP cellular uptake was evaluated after incubation with IONPs ([Fe] = 50 μg·mL^−1^). Perls’ Prussian blue colorimetric assay [35] was used to evaluate the uptake by A549 cells after 6 h, 24 h and 48 h of incubation. 

As illustrated in Figure 2A,B, while similar iron quantities were observed after 6h for both batches, the carboxylated nanoparticles showed significantly higher uptake than the PEGylated ones at both 24 and 48 h (approximately 3 and 10 times higher, respectively). A significant decrease in the internalization pattern was observed between 24 h and 48 h for both formulations.

### 3.3. Particle Stability 

Owing to their surface composition and their colloidal stability in culture media, it was expected that the two kinds of nanoparticles would exhibit different uptake behaviors [41]. The stability of both formulations in the considered media may explain the resulting differences more than their charges. Contrary to PEGylated particles, PCS measurements for the carboxylated IONPs in MEM indicated an increase in the hydrodynamic diameter over time (Appendix A), which could be attributable to the poor stability of negatively charged particles in the medium or to the physisorption of organic macromolecules. One method to prove the colloidal stability of magnetic nanoparticles in a given medium consists of measuring the evolution of their longitudinal (r_1_) and transverse (r_2_) relaxivities when diluted in that media [42]. The clustering of several magnetic cores usually leads to an increase in the r_2_/r_1_ ratio that can reach a value of up to several hundred at high magnetic fields (Larmor frequency). Contrary to PEGylated particles, carboxylated IONPs showed a significant increase in the relaxivity ratio with time (Appendix A), thus indicating a slow destabilization of the system. However, when comparing this evolution to the one observed in the absence of proteins within the medium, i.e., without FBS, it is interesting to note that despite the increase in the relaxivity ratio, the suspension remained stable for at least 48h, whereas a complete sedimentation was observed after 1h in the medium free of proteins (Appendix A). Taken together, these data suggest the formation of a “stabilizing” protein crown surrounding the carboxylated particles. In the case of PEGylated particles, neither a destabilization nor a protein crown formation could be observed.

### 3.4. Confocal Microscopy

The intracellular localization of IONPs and their spatial distribution were investigated by confocal microscopy. Detection was possible using rhodamine labeled IONPs. The cells incubated in the presence of carboxylated IONPs for 2 h already exhibited some red spots, highlighting that IONP aggregates were mainly distributed in the close vicinity of the plasma membrane. When increasing the incubation time, the number of IONP aggregates increased and the clusters moved from regions near the plasma membrane to the perinuclear region over time. No clusters were detected within the nucleus (Appendix A). By using a lysotracker, a high co-localization percentage (Appendix A) with lysosomes was observed regardless of the incubation time (Appendix A). In the case of PEGylated IONPs (Appendix A), the number of red spots was significantly lower than carboxylated IONPs, suggesting that the particles enter the cells as individual objects (or small aggregates) (Figure 3).

### 3.5. X-ray Irradiation

To determine the potential increase in cell damage induced by the presence of IONPs, 225 kV X-ray irradiations were administrated to cells pre-incubated with and without particles over time (6 h, 24 h and 48 h). The cell survival fraction at 2 Gy was determined by clonogenic assays (Figure 4). With the exception of the shortest incubation time, a significant decrease in the survival fractions was observed for both types of NPs (*p* < 0.01).

The amplification factors (AF; Table 1) were calculated to quantify the ability of IONPs to enhance cell death. Whatever the type of NPs or incubation time, no correlation between the maximum AF and cellular IONPs uptake was observed (r = −0.2000 by Spearman’s analysis; *p*–value: 0.7). This observation appears more striking when comparing IONPs-PEG_5000_ after 6h and 48h of incubation, for which AF values were doubled, whereas the quantity of total iron was significantly lower (by approximatively one-third). 

Physical calculations of the dose enhancement induced by X-ray/IONP interaction were carried out. Owing to their size (i.e., 7 nm) and the preparation method, we hypothesized that maghemite constituted the main phase of the as-prepared IONPs [43,44]. Based on this, Appendix A shows an increase in the attenuation cross-section when X-rays pass through Fe_2_O_3_ depending on the concentration considered. The ratio of the attenuation cross-section between Fe_2_O_3_ and water demonstrated that Fe_2_O_3_ can absorb approximately 25 times more photon energy than water (Appendix A). In comparison, gold nanoparticles can have an efficiency of 100 times higher than water in this context [29]. From these data, a maximum physical enhancement of 0.23 DEU/WP of Fe_2_O_3_ has been calculated, as illustrated in Appendix A. It should be noted that an enhancement of 1 DEU reflects a doubling of the deposited dose in the presence of the considered radio-enhancer. The results thus indicate that the calculation of the predicted theoretical dose induced by the interaction of ionizing radiation with iron oxide nanoparticles is relatively low.

### 3.6. TrxR Activity upon IONPs Treatment

According to recent reports focusing on gold NPs (GNPs), some authors have identified the thioredoxin reductase (TrxR) enzyme as a new potential biological target of metal-based NPs in both cancer [28,29,30] and healthy [45] cell lines. In these models, the authors deomstrated that metallic ions can be released due to the acidic pH of lysosomes after the uptake, exerting their action through an interaction with the thiol and selenol groups of the TrxR enzyme. Therefore, IONP degradation kinetic profiles were determined in ALF for both kinds of nanoparticles according to a procedure previously described [34] (Appendix A). It is interesting to note that most of the nanoparticles were digested after 48 h of incubation. On the other hand, it appeared that the particles degraded more slowly when stabilized by an additional organic layer, with PEG acting as a protective coating against the acidic environment. No dissolution of IONPs was observed in SBF-containing medium for both coating compositions.

In this context, and regarding the dissolution properties of these IONPs in acidic media, we were interested in determining if the two types of NPs could modulate the activity of TrxR over time (6 h, 24 h and 48 h). Figure 5 shows the TrxR activity in A549 cells incubated with ([Fe] = 50 μg·mL^−1^) and without IONPs. A decrease in the TrxR activity was observed for both types of NPs regardless of the incubation time. The magnitude of TrxR inhibition seemed to increase over the incubation time for A549 cells pre-incubated with carboxylated IONPs, while a plateau appeared from 24 h for A549 cells pre-incubated with PEGylated IONPs. 

Similar to the irradiation results, no correlation between the amount of internalized iron and the TrxR residual activity could be highlighted for both formulations (r = 0.6446 by Pearson’s analysis; *p*-value: 0.17). For the carboxylated particles, iron content decreased by half after 24 h (Table 1), whereas the percentage of TrxR inhibition reached a maximum at 48 h (Figure 5). On the other hand, a plateau for the percentage of TrxR inhibition between 24 h and 48 h was observed for the PEGylated particles while iron content was almost 10 times lower at 48 h compared to 24 h.

More interestingly, when plotting the amplification factor at 2 Gy vs. the residual TrxR activity level, a high degree of correlation can be noted (r = −0.913 by Pearson’s analysis; *p*-value: 0.0023) as already reported in the case of GNPs [38] (Figure 6). 

## 4. Discussion 

Owing to the fact that larger particles (typically above 50 nm) could be preferentially trapped by the reticuloendothelial system following intravenous injection (leading to fast accumulation in the liver and spleen), we focused on the preparation of sub-10 nm IONPs in this study [46]. These particles were obtained by the preparation of magnetic cores by alkaline co-precipitation of ferrous/ferric chlorides in polyol media, followed by a surface modification using a silanization compound (TEPSA). The benefits of using this agent lie in the stability of the deposited coating (formation of a stable Fe–O–Si covalent bond) and in the relative control of the layer’s thickness. By proceeding this way, 7 nm IONPs were obtained with a good control over their size (PDI: 1.2) and shape (mostly spherical). The analysis of TEPSA-treated particles by PCS revealed monomodal size distributions and suggested the coating of individual objects with no aggregates (Figure 1E). Recent studies [32] have demonstrated the importance of chain length on PEGylated IONP biodistribution. More specifically, IONPs bearing PEG_5000_ showed significantly longer circulating times in comparison with IONPs bearing a lower molecular weight PEG_800_. Despite the uncertainty related to the existence of an EPR effect in solid tumors (especially when comparing humans to murine tumor models), it can be agreed that increased circulation time is a prerequisite in promoting the uptake and accumulation of IONPs within tumors [47]. In the present work, TEPSA-IONPs PEGylation was achieved by classical EDC coupling process using α-amino and ω-methoxy-PEG, and assessed by PCS and zeta potential measurements. 

In accordance with other reports [39,40], the IONPs prepared for this study had a negligible influence on the cell viability of A549 cells regardless of the incubation time or coating material. Owing to their surface composition and their respective colloidal stability in culture media [41], it was expected that the two types of IONPs would exhibit different uptake behaviors. Internalization studies confirmed this hypothesis. Although similar amounts of iron were quantified after a 6h incubation, significant differences in uptake yields were observed for longer incubation times. This observation suggests different intracellular trafficking and fates depending on their initial surface composition, which in turn influences the NP stability or their interaction with their environment. As an example, it is commonly accepted that proteins contained in the medium preferably bind to negatively or positively charged particles in comparison with neutral ones [48,49,50]. These adsorbed proteins can influence the interaction with cells by promoting the receptor-mediated endocytosis, thus affecting their uptake mechanisms [51]. In this study, the formation of a protein layer surrounding carboxylated IONPs clusters has been highlighted by their increased stability in FBS-containing medium (vs. total sedimentation after 1h without FBS) (Appendix A). On the other hand, no protein adsorption could be highlighted for PEGylated IONPs, suggesting a possible alternative uptake mechanism.

Although the benefit of the use of high-Z NPs has been widely demonstrated in in vitro and in vivo studies, all of these nanomaterials have yet to receive FDA approval for clinical use. As discussed in [9,29], the debate around the basic mechanism of nanoparticle radiosensitization is one of the hurdles limiting the clinical translation of this technology. Initially, high-Z nanomaterials were classified as radioenhancer agents due to their higher mass energy absorption coefficient compared to soft tissue. The observed effect was attributed to an increase in the locally deposited dose originating from the emission of secondary particles after interaction between the nano-object and the photon beam. 

In this study, we demonstrated a significant radiosensitizing effect of both IONP formulations in combination with 225 kV X-rays photons. Interestingly, the observed X-ray effect does not seem to be correlated to the iron content in A549 cancer cells, contrary to what is predicted in the above-mentioned theory. Furthermore, a non-statistically significant 1.6% increase in the maximum dose enhancement (0.23 DEU/WP; Appendix A) was calculated based on X-ray attenuation cross-sections in the case of IONP PEG_5000_ incubated for 24 h (1.6 pg/cell), whereas our in vitro results showed a SER_10%_ of 1.10 ± 0.04 under the same experimental conditions (Appendix A). Although theoretical considerations do not predict a significant macroscopic dose enhancement in our experimental design, models of metallic NP-radiation interactions demonstrated a highly heterogeneous dose distribution at the nanoscale [52,53]. This nanoscopic dose distribution is characterized by very intense and localized energy deposition in the immediate vicinity of the nano-objects. We cannot, therefore, exclude that these nanoscopic mechanisms, widely described in the context of gold NPs, occur with IONPs. However, the local dose increase is itself dependent on the difference in the respective attenuation coefficients of the materials used. A small increase in nanoscopic dose can therefore be expected in the presence of IONPs due to their low attenuation coefficient. Overall, this highlights that the proposed radioenhancing mechanism of action cannot solely explain the observed in vitro results.

A mathematical analysis of our in vitro results shows an increase in the radiobiological α/β ratio (Appendix A), as already reported for other nano-objects [29,54], suggesting that cells containing IONPs have a lower ability to repair DNA damage produced by irradiation. This opens the door to a potential role of biology in the mechanism of action [55,56]. In a recent study [28], we suggested that gold nanoparticles exert a radiosensitizer role by weakening some key biological pathways in the cells before irradiation. In this theory, the internalization of nanoparticles in lysosomes triggers the release of metal ions that can inhibit enzymes from the thioredoxin reductase family, preventing the regeneration of intracellular antioxidants and affecting the management of oxidative stress. We demonstrated that this enzymatic inhibition could lead to a drop in ATP content that interferes with various pathways including DNA repair pathways post-irradiation. Poor or incomplete DNA repair can then induce cell death [57]. The results obtained in the present study seem to corroborate this hypothesis, suggesting that IONPs can act in a similar way to gold nanoparticles. Confocal microscopy results show a high colocalization of signals associated with nanoparticles and lysosomes (>80% independent of incubation time or IONP formulation; Appendix A), suggesting an internalization of the nano-objects in these vesicles. The kinetic measurements of IONP degradation in an acidic medium showed an increase in iron ions released over time (Appendix A). Moreover, Fe^2+^/Fe^3+^ have been identified as reagents able to oxidize thiol and selenol groups which constitute the active sites of the thioredoxin reductase family enzymes. This could explain the subsequent decrease in the activity of TrxR that we reported under all experimental conditions (Figure 5). Interestingly, no correlation between intracellular iron content and the TrxR residual activity has been highlighted. For both types of NPs, iron content decreased after 24h of incubation, whereas the level of TrxR inhibition reached a maximum after 48h of incubation. Moreover, we reported that the activity level of TrxR after IONP incubation is inversely correlated to the magnitude of the radiosensitizing effect, as already observed in the case of gold nanoparticles [38]. The inhibition of these enzymes could thus limit the capacity of the cells to counteract the significant increase in intracellular ROS produced following irradiation and/or by IONPs through the Fenton and Haber–Weiss reactions [18,58]. Furthermore, the catalytic surface properties of IONPs under X-ray irradiation have been described as having a synergetic effect on Fe^2+^/Fe^3+^ release by Klein et al. [17,58]. In this study, the ROS concentration is approximately two times more important in X-ray exposed MCF-7 cells in the presence of citrate SPIONs compared to Fe^2+^ incubated MCF-7 cells or to non-irradiated MCF-7 cells exposed to citrate SPIONs.

Overall, these results highlight the importance of biological phenomena in the radiosensitizing effect induced by NPs and the key role played by TrxR in the mechanism. This outcome opens the door to TrxR activity measurements as a screening test for the improvement of nanoparticles used in radiation therapy applications.

## 5. Conclusions

With the emergence of MRI-guided radiotherapy devices such as MR-LINAC [59], the development of platforms exhibiting both radiosensitization and contrast agent properties is vital. In this context, the study and use of IONPs seem promising due to their wide use in the biomedical field. This study describes the synthesis and the characterization of two IONP formulations with the same magnetic core size (around 7 nm), but with two different coatings (i.e., PEG_5000_ and carboxylic acids). After determination of their cytotoxicity and internalization behavior, their radiosensitizing potential was assessed in vitro on lung adenocarcinoma cells using 225 kV X-ray photons. Our results demonstrate a significant radiosensitizing effect of both IONPs formulations on A549 cells and confirmed some recent hypotheses suggesting that NPs radiosensitize cells by weakening them through the inhibition of key antioxidant enzymes.

It appears interesting to note that no correlation between the total amount of internalized iron and the residual TrxR activity has been evidenced. This observation, which seems counter-intuitive at first sight, could be explained by considering the different intracellular fates of the different IONP formulations. This particular aspect illustrates the importance of the physicochemical properties of IONPs in the understanding of the mechanisms related to their radiosensitization properties and underlines the importance of a rigorous characterization of formulations.

Finally, it seems interesting to note that PEGylated NPs exhibit a significant radiosensitization effect with an SER_10%_ of 1.1 (in the same experimental conditions, an SER_10%_ of 1.22 was calculated for PEGylated GNP [28]). According to some preliminary data, the observations made for X-ray irradiations can be extended to proton irradiations (Appendix A). Owing to the great stability of these nanoobjects and their capacity to circulate for a prolonged time within the bloodstream following an IV injection, these results appear to be particularly exciting and promising for the treatment of solid tumors. In this context, the use of active tumor targeting strategies appears vital to circumvent the limitations of the EPR model.

## Figures and Tables

**Figure 1 nanomaterials-13-00201-f001:**
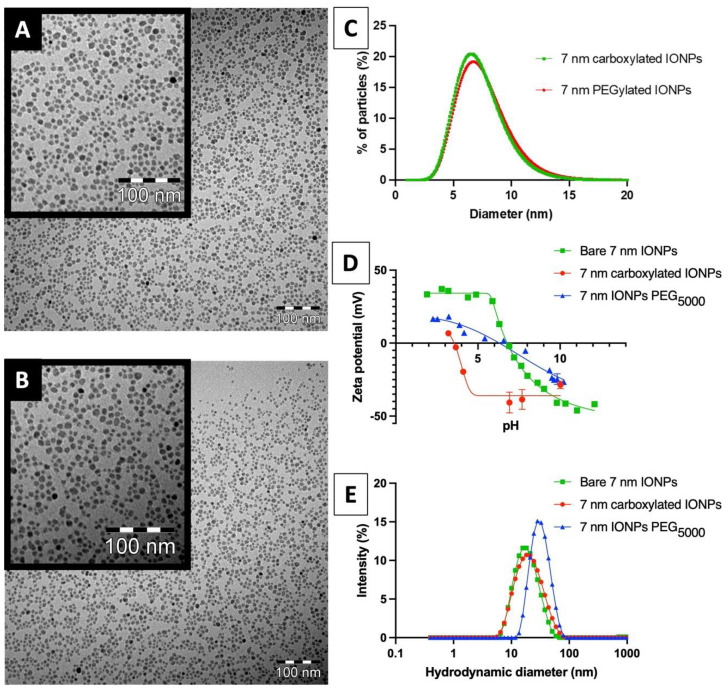
(**A**) TEM image of 7 nm carboxylated IONPs; (**B**) TEM image of 7 nm PEGylated IONPs, the scale bar corresponds to 100 nm. (**C**) Comparison of size distributions (histograms fitted with a log-normal function), obtained after statistical analysis of TEM images; (**D**) zeta potential measurements of each sample obtained by PCS; (**E**) size distributions of each sample obtained by PCS.

**Figure 2 nanomaterials-13-00201-f002:**
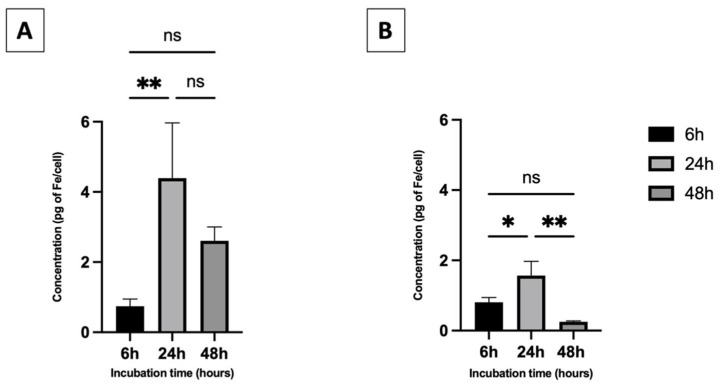
Iron concentrations in A549 cells assessed after 6 h, 24 h and 48 h of incubation with (**A**) 7 nm carboxylated IONPs and (**B**) 7 nm IONPs PEG_5000_ ([Fe] = 50 µg·mL^−1^). Cellular iron content was quantified by the Perls’ Prussian blue stain method. Iron concentrations are expressed as mean values ± S.D for three independent experiments (Tukey test, ** *p* < 0.01, * *p* < 0.05, ns = not significant).

**Figure 3 nanomaterials-13-00201-f003:**
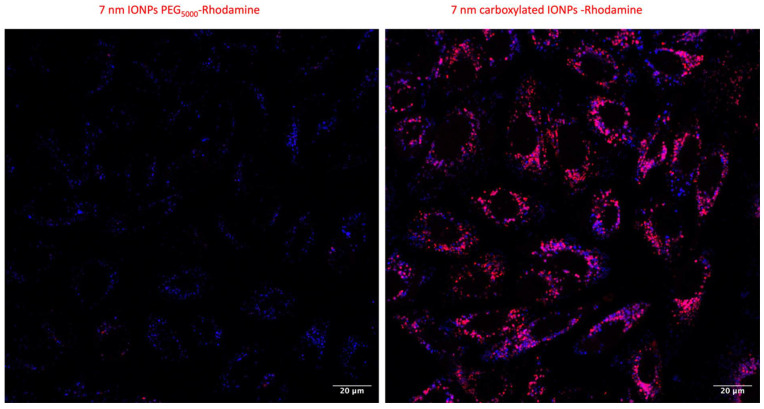
Confocal microscopy micrographs of iron oxide nanoparticle uptake in A549 cells and intracellular localization. Cells preincubated with 7 nm IONPs PEG_5000_ (**left**); 7 nm carboxylated IONPs (**right**) ([Fe] = 50 µg·mL^−1^) for 48 h (scale bar: 20 μm). Red channel: Rhodamine-IONPs and Blue channel: LysoTracker Blue.

**Figure 4 nanomaterials-13-00201-f004:**
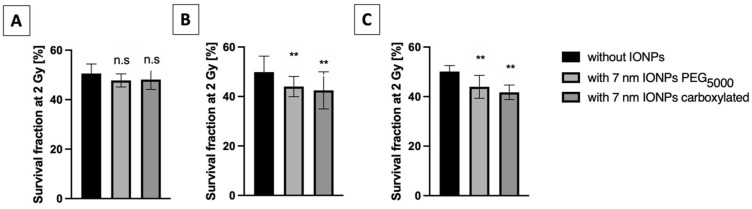
Survival fractions determined by standard clonogenic assay for A549 cells pre-incubated with and without IONPs ([Fe] = 50 μg·mL^−1^) for (**A**) 6 h, (**B**) 24 h and (**C**) 48 h and irradiated by X-rays. Data are plotted as means ± SD from three independent experiments. One-way ANOVA analysis was performed for each result (Tukey test, ** *p* < 0.01, ns = not significant).

**Figure 5 nanomaterials-13-00201-f005:**
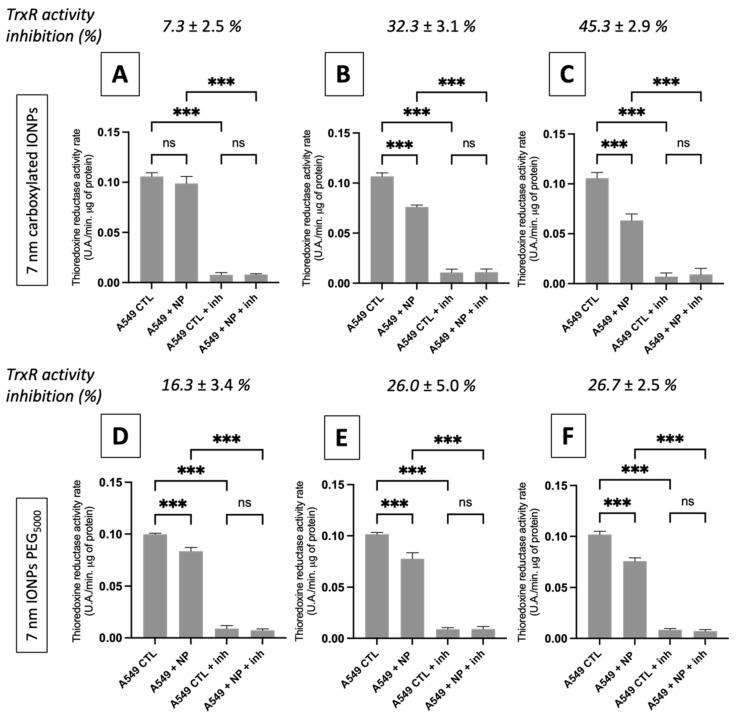
TrxR activity rates calculated from the slope of corresponding TrxR activity curves extracted from the measurement of absorption at 412 nm during 10 min for A549 cells treated with ([Fe] = 50 µg·mL^−1^) and without 7 nm carboxylated IONPs and 7 nm IONPs PEG_5000_ for 6 h of incubation (**A**,**D**), 24 h of incubation (**B**,**E**) and 48 h of incubation (**C**,**F**). Data are plotted as means ± SD from three independent experiments. One-way ANOVA analysis was performed for each result (Tukey test, *** *p* < 0.001, ns = not significant).

**Figure 6 nanomaterials-13-00201-f006:**
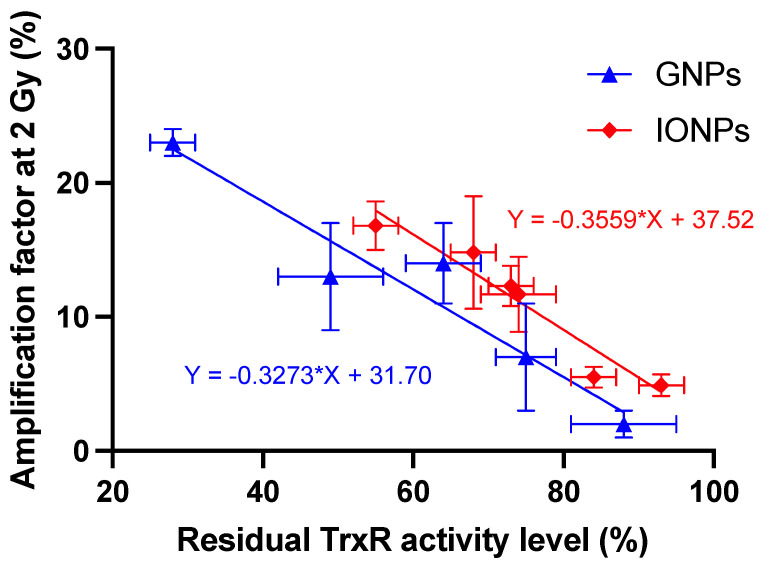
The amplification factor at 2 Gy obtained in A549 cells as a function of residual TrxR activity. Gold data (blue) reported by Penninckx et al. [38] showed the influence of 10 nm amino-PEG GNPs on five different cell lines, while IONP data (red) showed the influence of two IONPs on A549 cells over time. Data are plotted as means ± SD from three independent experiments. Pearson’s correlation analysis was carried out for the scatterplot (r = −0.913).

**Table 1 nanomaterials-13-00201-t001:** Concentrations of iron in cells and amplification factor for A549 cells pre-incubated for 6 h, 24 h or 48 h with 7 nm carboxylated IONPs and 7 nm IONPs PEG_5000_ ([Fe] = 50 µg·mL^−1^) before being irradiated with 2 Gy X-rays.

	Incubation Time (hours)	Internalization in pg of Fe/cell	AF at 2 Gy (%)
IONPs PEG_5000_	6 h	0.8 ± 0.1	5.5 ± 0.8%
	24 h	1.6 ± 0.4	11.7 ± 2.8%
	48 h	0.25 ± 0.03	12.3 ± 1.5%
Carboxylated IONPs	6 h	0.7 ± 0.2	4.9 ± 0.8%
	24 h	4.6 ± 1.2	14.8 ± 4.2%
	48 h	2.6 ± 0.4	±1.8%

## Data Availability

Not applicable.

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
