# Peer review of "Advances in the Mechanistic Understanding of Iron Oxide Nanoparticles’ Radiosensitizing Properties"

_nanomaterials, 2023, doi:10.3390/nano13010201_

Round 1

Reviewer 1 Report

Dear Authors and Dear Editor, 

In the manuscript entitled “Advances in the mechanistic understanding of iron oxide nanoparticle radiosensitizing properties” Stanicki and co-workers extend previous studies on the radiosensitizing properties of gold nanoparticles to carboxylated and pegylated IONPs. The authors suggest that the enhancement of the radiosensitizing properties of their IONPs results, in addition to physical phenomena, also from cell weakening due to inhibition of ROS-detoxifying antioxidant enzymes, particularly thioredoxin reductase, by metal ions (Fe2+/3+) leached during lysosomal internalization of the IONPs. The authors propose that this may be a general mechanism, common to metal nanoparticles.

The study is well planned and executed to high technical and scientific standards. The experimental procedures are described in enough detail to warrant reproduction by interested researchers. The results are well presented and meaningfully discussed.

This study is likely to prompt other researchers to study further the complex interplay between nanoparticles´ physical-chemical, biological and radiosensitizing properties, eventually leading to the rational design of nanoparticles with optimized radiosensitizing properties.

In my opinion the manuscript is scientifically sound and ready for publication.

Author Response

Authors would like to thank the reviewer 1 for his/her constructive feedback on this research article. His/Her comments were highly appreciated, and the authors hope to meet his/her expectations after the revision. 

Reviewer 2 Report

The work presented by the authors address a hot topic in radiotherapy research, namely the enhancement of radiotherapy yield by combination with nanoparticle. The IONPs were used to find out a radiosensitization effect on A549 cell cultures. In spite of very exciting results (the corelation of irradiation effect with the inhibition of TrxR Activity) the data presentation is very unappropriated (some data are missing (proton irradiation results), some important data are placed in SM, and any way the SM is not present within the manuscript). Detailed observations are inserted as comments in the manuscript (attached file).

Author Response

please see document

Reviewer 3 Report

This study demonstrates the radiosensitizing effect of iron oxide nanoparticles capped by two different coatings (carboxylated and pegylated IONPs) and corroborates the hypothesis that nanoparticles exhibit a radiosensitizing effect by weakening cells through the inhibition of detoxification enzymes. The manuscript contains original (unpublished before) and useful information which may be interesting to some of the Nanomaterials readers. However, the manuscript needs major revisions and improvement with regard to the proper interpretation of the findings. Throughout the ideas and concepts are frequently presented without sufficient grounding, with speculative interpretations presented as fact (or accepted as generally true) and without citation. Additionally, the paper suffers from minor grammatical errors throughout. Therefore, the submission may be considered for possible publication only after addressing the following comments:

1. The abstract needs improvement. Please add more information on the research results. It should be informative and completely self-explanatory and should not include experimental procedure.

2. Introduction should be rewritten by including more recent references/literature as many papers have been published recently on the radiosensitizing properties of iron oxide nanoparticles.

3. The novelty of the research should be clearly described at the end of the introduction part.

4. Material and Methods: The methods used for the synthesis of iron oxide nanoparticles should be described in more details.

5. “iron oxide nanoparticle” should be replaced with “iron oxide nanoparticles” throughout the manuscript.

6. XRD measurements: The particles sizes should be calculated by Debye Scherrer formula.

7. The quality of figures is low and should be improved.

Scientific Reports 12(1) (2022), 9602. https://doi.org/10.1038/s41598-022-13368-x

Journal of Materials Chemistry B 9(22) (2021), 4510-4522. 10.1039/D0TB02561E

New Journal of Chemistry 40(6), 5221-5230. DOI: https://doi.org/10.1039/C5NJ03594E

Materials Science and Engineering: C 129 (2021), 112394. https://doi.org/10.1016/j.msec.2021.112394

ACS Applied Materials & Interfaces, 13 (2021) 13072-13086, https://doi.org/10.1021/acsami.0c21076

Nosrati, H., Baghdadchi, Y., Abbasi, R., Barsbay, M., Ghaffarlou, M., Abhari, F., Mohammadi, A., Kavetskyy, T., Bochani, S., Rezaeejam, H., Davaran, S. and Danafar, H.  2021.  Iron oxide and gold bimetallic radiosensitizers for synchronous tumor chemoradiation therapy in 4T1 breast cancer murine model. Journal of Materials Chemistry B 9(22), 4510-4522.

Rahman, W.N., Kadian, S.N.M., Ab Rashid, R., Abdullah, R., Abdul Razak, K., Pham, B.T.T., Hawkett, B.S. and Geso, M.  2019.  Radiosensitization characteristic of superparamagnetic iron oxide nanoparticles in electron beam radiotherapy and brachytherapy. Journal of Physics: Conference Series 1248(1), 012068.

Sood, A., Dev, A., Sardoiwala, M.N., Choudhury, S.R., Chaturvedi, S., Mishra, A.K. and Karmakar, S.  2021.  Alpha-ketoglutarate decorated iron oxide-gold core-shell nanoparticles for active mitochondrial targeting and radiosensitization enhancement in hepatocellular carcinoma. Materials Science and Engineering: C 129, 112394.

8. FT-IR analysis should be performed to detect the functional groups of the nanoparticles.

9. The conclusion part should be improved. It should contain the key message that has been discussed in the manuscript. It should be written in an effective manner and should present an understanding of the whole story.

10. Elemental composition of the samples should be studied quantitatively.

11. Please number consecutively any equations that have to be displayed in the text.

Author Response

please see document

Round 2

Reviewer 3 Report

  The authors have addressed my comments and concerns in the revised version. Overall, the manuscript is well-written and the reported results are of valuable interest to readers. However, the following references should be cited in the introduction part before publication to improve the quality of the manuscript: 13:33  

Scientific Reports 12(1) (2022), 9602. https://doi.org/10.1038/s41598-022-13368-x

Journal of Materials Chemistry B 9(22) (2021), 4510-4522. 10.1039/D0TB02561E

New Journal of Chemistry 40(6), 5221-5230. DOI: https://doi.org/10.1039/C5NJ03594E

Materials Science and Engineering: C 129 (2021), 112394. https://doi.org/10.1016/j.msec.2021.112394

ACS Applied Materials & Interfaces, 13 (2021) 13072-13086, https://doi.org/10.1021/acsami.0c21076